# Correlation of a commercial platform's results with post-vaccination SARS-CoV-2 neutralizing antibody response and clinical host factors

**Rebecca Slotkin**[1], **Tassos C. Kyriakides**[1,2,3], **Anupam Kundu**[2,3], **Gary Stack**[1,3], **Richard E. Sutton**[1,3], **Shaili Gupta**[1,3]*

**1** Yale School of Medicine, New Haven, Connecticut, United States of America, **2** Yale Center for Analytical Sciences, Yale School of Public Health, New Haven, Connecticut, United States of America, **3** Veterans Healthcare System of Connecticut, West Haven, Connecticut, United States of America

* Shaili.gupta@yale.edu

## Abstract

### Introduction

The objective of this study was to describe the correlation between the commercially available assay for anti-S1/RBD IgG and protective serum neutralizing antibodies (nAb) against SARS-CoV-2 in an adult population after SARS-CoV-2 vaccination, and determine if clinical variables impact this correlation.

### Methods

We measured IgG anti-S1/RBD using the IgG-II CMIA assay and nAb $IC_{50}$ values against SARS-CoV-2 WA-1 in sera serially collected post-mRNA vaccination in veterans and healthcare workers of the Veterans Affairs Connecticut Healthcare System (VACHS) between December 2020 and January 2022. The correlation between IgG and $IC_{50}$ was measured using Pearson correlation. Clinical variables (age, sex, race, ethnicity, prior COVID infection defined by RT-PCR, history of malignancy, estimated glomerular filtration rate (GFR calculated using CKD-EPI equation) were collected by manual chart review. The impact of these clinical variables on the IgG-nAb correlation was analyzed first with univariable regression. Variables with a significance of $p < 0.15$ were analyzed with forward stepwise regression analysis.

### Results

From 127 sera samples in 100 unique subjects (age 20–93 years; mean 63.83; SD 15.63; 29% female; 67% White), we found a robust correlation between IgG anti-S1/RBD and nAb $IC_{50}$ ($R^2 = 0.83$, $R^2_{adj} = 0.70$, $p < 0.0001$). Race, ethnicity, and a history of malignancy were not significant on univariable analysis. GFR ($p < 0.05$) and prior COVID infection ($p < 0.001$) had a significant impact on the correlation between IgG anti-S1/RBD and nAb $IC_{50}$. Age ($p = 0.06$) and sex ($p = 0.07$) trended towards significance on univariable analysis, but were not significant on multivariable regression.

**Data Availability Statement:** De-identified individual participant-level data that underlie the results reported in this paper are shared in S1 File.

**Funding:** Funding for this study was provided by the National Institute of Allergy and Infectious Diseases, National Institutes of Health (R01 AI150334 to RES). The analysis was supported by Global Health Equity Scholars through the Fogarty International Center of the National Institutes of Health and the National Institute of Arthritis and Musculoskeletal and Skin Diseases of the National Institutes of Health (FIC D43TW010540 and T32AR007107 to RS). The funders had no role in the study design, data collection and analysis, decision to publish, or preparation of the manuscript.

**Competing interests:** The authors have declared that no competing interests exist.

## Conclusions

There was a strong correlation between IgG anti-S1/RBD and nAb $IC_{50}$ after SARS-CoV-2 vaccination. Clinical comorbidities, such as prior COVID infection and renal function, impacted this correlation. These results may assist the prediction of post-vaccination immune protection in clinical settings using cost-effective commercial platforms.

## Introduction

High levels of anti-SARS-CoV-2 neutralizing antibodies (nAb) have been shown to correlate with protection against severe infection with SARS-CoV-2 [1]. Messenger RNA (mRNA) vaccines elicit the production of nAb, but the ability to reliably detect nAb is limited to research laboratory settings. The FDA is in the process of deciding the optimal vaccination schedule for SARS-CoV-2 with a goal of simplifying the vaccination regimen [2]. Neither the CDC nor FDA currently recommend using commercially available antibody testing to assess post-vaccination immunity or to determine the timing of booster doses [3,4]. However, using commercially available assays that reliably predict nAb levels in clinical settings could allow for rapid and customizable decisions for patients regarding the optimal SARS-CoV-2 vaccination schedule and timing of booster doses.

Although no immunoassay has yet been thoroughly validated for post-vaccination serological testing, the Abbott Architect chemiluminescent microparticle immunoassay for SARS-CoV-2 IgG anti-S1/RBD (IgG-II CMIA) was used to study serological response after mRNA vaccination [5]. According to the manufacturer, values of >590.72 BAU/mL (>4160 AU/mL) correspond to a 95% probability (95% confidence interval [CI]: 78%–99%) of neutralization capacity against WA-1 [6]. The correlation of this assay's results with the neutralizing capability of sera post-vaccination and the impact of patient clinical factors on this correlation have yet to be fully explored. The goal of this study is to measure the correlation between the SARS-CoV-2 IgG anti-S1/RBD and the sera neutralizing antibodies in a vaccinated US adult population, taking into account prior COVID-19 infection and individual clinical factors which may impact immune function.

## Methods

### Participants

Approval for this study (SARS-CoV-2 Vaccine and You, Protocol #1599488) was obtained from the Veterans Affairs Connecticut Healthcare System (VACHS) Institutional Review Board (IRB) in December 2020. Written informed consent was obtained from willing participants recruited from COVID-vaccination clinics set up by VACHS beginning in December 2020. Participants were eligible if they were either veterans or healthcare workers working at VACHS. Participants were excluded if they had an illness or condition that prohibited their informed consent.

### Variable collection

We measured IgG anti-S1/RBD values on the CMIA assay and sera neutralizing antibody $IC_{50}$ values against pseudotyped WA-1 on sera collected from willing participants at standard intervals post-mRNA vaccination as previously described (pre-dose-1, pre-dose 2, post-dose-2 at 1, 3, 6, 12 months, with an additional collection at 1-month post-dose-3, approximately 10 months post-dose 2) [7]. Each measurement of IgG and $IC_{50}$ was done on serum from the same collection time point to avoid any errors in measurement due to antibody fluctuation

over time. IgG anti-S1/RBD were measured in AU/ml, a unit specific to the commercial platform used. One Abbott platform AU/ml converts to 7.04 BAU/ml, the standard World Health Organization (WHO) units used as a standard for commercial antibody platforms [8,9]. Clinical variables were collected by manual chart review as previously described [7]. See S1 File for the de-identified data file.

## Statistical analysis

We calculated the correlation between $\log_{10}$IgG and $\log_{10}$IC$_{50}$ on all sera samples. The correlation was measured using a linear correlation adjusted and un-adjusted for the number of variables. Values read as >25,000 on the commercial platform were given the value 25,000. Three samples with missing IC$_{50}$ values were not included in the analysis. We performed a sensitivity analysis examining the correlation without any values that were above the limit of detection (read as >25,000) on the commercial platform (S1 Fig).

For the analysis of impact of clinical variables, only unique subject samples were included, and values measured from other timepoints in the same subject were removed. There were 27 subjects who had samples collected at multiple time-points post mRNA-vaccination. For these 27 subjects, each subject's latest available serum collection time-point was selected for the analysis of the impact of clinical variables. COVID-positivity was defined as RT-PCR COVID-diagnosis at any time prior to collection (S2 Fig). As there were not enough samples at each time-point for a time-based subgroup analysis, this was not performed.

We examined the impact of clinical factors [age, sex, race, ethnicity, prior COVID infection defined by RT-PCR, history of cancer (malignancy), estimated glomerular filtration rate (GFR calculated using CKD-Epi equation)] on the relationship between $\log_{10}$IgG and $\log_{10}$IC$_{50}$ using regression analysis. Each individual clinical factor was evaluated in a regression model with $\log_{10}$IgG to predict $\log_{10}$IC$_{50}$. All factors that were significant ($p < 0.15$) on this univariable analysis were further analyzed with multivariable regression. All the clinical variables that were significant ($p < 0.05$) on forward stepwise multivariable regression were used to create a $\log_{10}$IC$_{50}$ prediction model (Model 1). This model was then compared to a prediction model using $\log_{10}$IgG values alone (Model 2) to examine the utility of factoring in clinical variables when using IgG anti-S1/RBD to predict nAb IC$_{50}$.

## Results

We obtained 127 sera samples from 100 unique subjects (Table 1). $\log_{10}$IgG values demonstrated a strong correlation with $\log_{10}$IC$_{50}$ values ($R^2 = 0.83$, $R_{adj} = 0.70$, $p < 0.0001$) (Fig 1), and significantly predicted $\log_{10}$ IC$_{50}$ values (beta = 0.56, $p = 0$) on univariable analysis. We found that a IgG of $> 25,000$ predicted a robust IC$_{50}$ of $> 400$ in all participants. A sensitivity analysis demonstrated an improvement in the adjusted $R^2$ value of the correlation to 0.80, suggesting that the linear model in Fig 1 is the most conservative estimate of the true correlation between IgG and nAb IC$_{50.}$

The following clinical factors had a significant impact on this correlation: COVID-positive status (beta = 0.38, $p < 0.01$), and GFR 0–30 mL/min/1.73 m$^2$ (beta = 0.26, $p < 0.01$). Age ($p = 0.06$) and female sex ($p = 0.07$) trended towards significance on univariable analysis. Malignancy, race, and ethnicity were not significant predictors. $\log_{10}$IgG (beta = 0.54, $p < 0.01$), COVID-positive status (beta = 0.38, $p < 0.001$) and GFR 0–30 mL/min/1.73 m$^2$ (beta = 0.19, $p < 0.05$) remained significant predictors of $\log_{10}$IC$_{50}$ after adjusting for age and sex on multivariable analysis. Compared to using the $\log_{10}$IgG results alone, including COVID status and GFR with the $\log_{10}$IgG result provided a significantly better prediction model of $\log_{10}$ IC$_{50}$ (partial F = 11.20, $p < 0.0001$) (Table 2).

**Table 1. Cohort demographics and morbidities.**

| Variable | | N (%) |
|---|---|---|
| N | | 100 unique subjects* |
| Female | | 29 (29%) |
| Age, mean (SD, [range]) | | 63.83 (15.63, [20–93]) |
| Race | | |
| | White | 67 (67%) |
| | Black | 20 (20%) |
| | Other/not reported | 13 (13%) |
| Ethnicity | | |
| Non-Hispanic | | 87 (93%) |
| Hispanic | | 7 (7%) |
| COVID-Positive Samples | | 15 (12%) |
| Glomerular Filtration Rate | | |
| | >60 mL/min/1.73 m$^2$ | 73 (73%) |
| | 31–59 mL/min/1.73 m$^2$ | 16 (16%) |
| | <30 mL/min/1.73 m$^2$ | 11 (11%) |
| History of Malignancy | | 32 (32%) |

*100 unique subjects provided a total of 127 samples.

## Discussion

Our results demonstrate a robust correlation between IgG anti-SARS-CoV-2 S1/RBD and anti-SARS-CoV-2 neutralization antibodies, suggesting that IgG anti-S1/RBD could be used to predict neutralizing antibody immune response. These findings may open avenues for further

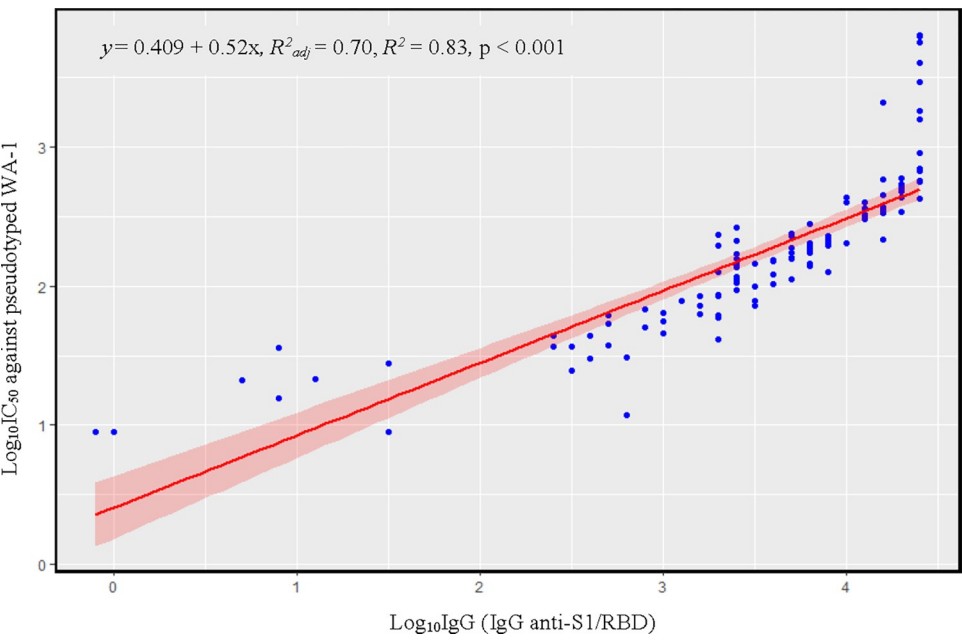

**Fig 1. Correlation between Log$_{10}$ IC$_{50}$ and Log$_{10}$IgG.** Illustrated above is the correlation between the log$_{10}$ IC$_{50}$ values against pseudotyped WA-1 and the log10 of serum IgG anti-S1/RBD value. IgG anti-S1/RBD values beyond the cut-off of 25,000 AU/mL (>log$_{10}$ = 4) were treated as 25,000 AU/mL. $R^2$ = 0.83, $R^2_{adj}$ = 0.70, p-value < 0.001.

**Table 2. Comparison of prediction models for $Log_{10}IC_{50}$ with and without clinical variables.**

|  | Model 1 | Model 2 |
|---|---|---|
| (Intercept) | 2.22 *** [2.16, 2.29] | 2.30 *** [2.24, 2.36] |
| $Log_{10}IgG$ | 0.44 *** [0.39, 0.49] | 0.46 *** [0.40, 0.52] |
| COVID positive | 0.37 *** [0.22, 0.52] |  |
| GFR 31–59 mL/min/1.73 m$^2$ | -0.02 [-0.16, 0.12] |  |
| GFR 0–30 mL/min/1.73 m$^2$ | 0.23 ** [0.06, 0.40] |  |
| N | 99 | 99 |
| $R^2$ | 0.78 | 0.70 |

**Table 2**. Only unique subjects (N = 100) were included the clinical variable regression analysis. The reference for glomerular filtration rate (GFR) was value > 60 mL/min/1.73 m$^2$, the reference for COVID status was COVID negative. All continuous predictors are mean-centered and scaled by 1 standard deviation. The table displays betas with confidence intervals in brackets for the significant clinical factors on multivariable analysis that were included in the model comparison. Model 1 includes clinical variables, model 2 has IgG alone. P-values indicated by stars

*** $p < 0.001$

** $p < 0.01$

* $p < 0.05$. Model 1 performed significantly better than model 2, partial F = 11.20, p < 0.0001. AIC for model 1 = 22.98, AIC for model 2 = 46.57.

use of commercially available antibody testing in lower-resourced research settings, where rapid diagnostic testing is generally preferred [10]. Antibody testing will not provide a guarantee against infection, but could help guide clinical decisions, such as the timing and frequency of re-vaccination with a goal of preventing severe infections. Antibody testing is already being used sporadically in primary care settings, although guidelines and evidence for this use are limited [11,12]. Further research into appropriate cut-off values which indicate immune protection, and the impact of host factors will be important to refine and better guide rapid antibody testing in clinical settings.

Immunity is a dynamic process dependent on individual clinical factors and a simplified yearly vaccination schedule may not be appropriate for everyone, potentially resulting in an unacceptably high rate of breakthrough infections over time in vulnerable individuals [13]. Factoring prior COVID-19 infection status and an estimation of renal function based on glomerular filtration rate when using IgG anti-S1/RBD strengthened the prediction of anti-SARS-CoV-2 neutralizing antibodies, while a history of malignancy and age did not significantly impact the correlation between nAb IC$_{50}$ and IgG anti-S1/RBD. This study highlights the importance of accounting for host clinical factors when analyzing immune response.

The neutralization antibody target in this study was WA-1. Several studies have suggested that post-vaccination neutralizing antibody response may be lower against the newer circulating omicron variants than WA-1 [14,15]. Given the robust correlation between WA-1 neutralizing antibody titers and anti-SARS-CoV-2 IgG, further research into how well the IgG correlates with immune protection against severe disease due to omicron variants is warranted. The clinical interpretation and application of this correlation may also be improved by updating commercial platforms with IgG against newer variants.

This study investigated a limited number of clinical factors in a vaccinated population and will need to be cautiously generalized to the broader population. Although we found that a prior history of malignancy, race, ethnicity, and sex were not a clinically significant factors in this study, we had a limited sample size and ethnic distribution. Larger correlation and modeling studies on populations with additional clinical factors and immunocompromising conditions will need to be conducted to better personalize the IgG anti-S1/RBD results.

## Conclusion

There is a robust correlation between sera IgG anti-S1/RBD and neutralizing antibody $IC_{50}$ titers, which is impacted by certain clinical variables such as prior COVID infection and renal function. Further understanding of the clinical factors that impact this correlation will be important for ongoing clinical or research applications of commercial antibody platforms. Commercially available antibody testing, such as the CMIA assay described in this study, can play an important role in lowering the cost and infrastructure barriers to assessing the levels of clinically active neutralizing antibodies. An understanding of neutralizing antibody titers may enable more nuanced patient conversations about SARS-CoV-2 immunity and frequency of vaccine booster doses, although further research will be needed to guide this use. We should leverage the tools we have available to refine rapid diagnostic antibody testing, lower barriers to SARS-CoV-2 research, and add clinical relevance to the optimal frequency of SARS-CoV-2 re-vaccination, especially in the presence of certain clinical factors.

## Supporting information

**S1 Fig. Additional correlation models.** (A) Loess Correlation Between $Log_{10}IC_{50}$ and $Log_{10}IgG$. The Loess (locally estimated scatterplot smoothing) is a non-linear correlation model which yields an adjusted $R^2$ of 0.84. We illustrate this model as a supplement because it does not produce a mathematical formula which would allow for easy replication and use, but does demonstrate the possibility of establishing an even stronger correlation for future applications. (B) Linear Correlation Between $Log_{10}IC_{50}$ and $Log_{10}IgG$ without IgG values over the detection limit. A sensitivity analysis examined the correlation if any values that were above the limit of detection (read as >25,000) on the commercial platform we used were removed. This sensitivity analysis demonstrated an improvement in the adjusted R-squared value of the correlation to 0.80.
(TIF)

**S2 Fig. Distribution of subjects by collection time-point and COVID-status.** COVID-positivity was defined as RT-PCR COVID- diagnosis at any time prior to collection.
(TIF)

**S1 File. Underlying data set for the study.**
(XLSX)

## Acknowledgments

We thank our laboratory staff who worked on our pseudotyping and commercial assay, as well as our subjects for donating their samples.

## Author Contributions

**Conceptualization:** Rebecca Slotkin, Shaili Gupta.

**Data curation:** Rebecca Slotkin, Gary Stack, Richard E. Sutton, Shaili Gupta.

**Formal analysis:** Rebecca Slotkin, Tassos C. Kyriakides, Anupam Kundu, Shaili Gupta.

**Investigation:** Rebecca Slotkin, Gary Stack, Richard E. Sutton, Shaili Gupta.

**Methodology:** Rebecca Slotkin, Tassos C. Kyriakides, Shaili Gupta.

**Project administration:** Shaili Gupta.

**Resources:** Gary Stack, Richard E. Sutton.

**Supervision:** Tassos C. Kyriakides, Shaili Gupta.

**Visualization:** Rebecca Slotkin, Anupam Kundu.

**Writing – original draft:** Rebecca Slotkin, Tassos C. Kyriakides, Gary Stack, Richard E. Sutton, Shaili Gupta.

**Writing – review & editing:** Rebecca Slotkin, Shaili Gupta.

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
