## [Decision Letter · Decision Letter 0]

27 Apr 2023

PONE-D-23-08631Correlation of a Commercial Platform’s Results with Post-vaccine SARS-CoV-2 Neutralizing Antibody Response and Clinical Host FactorsPLOS ONE

Dear Dr. Gupta,

Thank you for submitting your manuscript to PLOS ONE. After careful consideration, we feel that it has merit but does not fully meet PLOS ONE’s publication criteria as it currently stands. Therefore, we invite you to submit a revised version of the manuscript that addresses the points raised during the review process.

We look forward to receiving your revised manuscript.

Kind regards,

Harapan Harapan, MD, PhD

Academic Editor

PLOS ONE

Journal Requirements:

2.Thank you for stating the following financial disclosure: 

  "Funding for this study was provided by the National Institutes of Health (awards R01 AI150334 to RES). The analysis was supported by Global Health Equity Scholars (FIC D43TW010540 to RS)."

5. Please amend either the title on the online submission form (via Edit Submission) or the title in the manuscript so that they are identical.

Reviewers' comments:

Reviewer's Responses to Questions

**Comments to the Author**

1. Is the manuscript technically sound, and do the data support the conclusions?

Reviewer #1: Partly

Reviewer #2: Yes

2. Has the statistical analysis been performed appropriately and rigorously? 

Reviewer #1: Yes

Reviewer #2: Yes

3. Have the authors made all data underlying the findings in their manuscript fully available?

Reviewer #1: Yes

Reviewer #2: Yes

4. Is the manuscript presented in an intelligible fashion and written in standard English?

Reviewer #1: Yes

Reviewer #2: Yes

5. Review Comments to the Author

Reviewer #1: This manuscript is an interesting topic, however, the data is not new and the content is suitable for publishing as a letter or short communication that extended the result from the previous publications. I suggest adding more data and text to make it suitable for the original article.

Major concerns.

1. Please add the IRB/EC approval number to the Methods.

2. Why you over emphasised the CMIA technique of the Abbott AdviseDx SARS-CoV-2 IgG II?

The CMIA is only the technique, but the target of the antibody is more important than the technique of commercial diagnostic assays.

Suggest using "IgG anti-S1/RBD" or "IgG anti-RBD" instead of "IgG-II CMIA" to make it more specific.

See more: https://www.fda.gov/media/146371/download

3. Suggest creating a Demographic table including the clinical and COVID-19 vaccine used to delineate the participant's demographic data.

Minor concerns.

1. Methods, suggest being divided into subtitles, such as Participants, Immunologic assessment, and Statistical analysis.

2. Suggest adding the conversion factor of the Abbott AdviseDx to the introduction, the real result from this instrument report is in the AU/mL but BAU/mL is converted. Some readers may don't understand where the BAU/mL came from.

Comments.

1. Suggest using subscript of the "50" in the term "IC₅₀".

Reviewer #2: Reviewer Suggestions

1. As it is known, antibody titers spontaneously decrease over time, and also may increase due to COVID-19 or vaccination. For this reason, to evaluate the correlations of the neutralizing antibody IC50 measurement and the anti-RBD-S1-IgG (IgG) antibody measurement, they must be studied from the sera samples taken at the same time-point. In order to indicate that the study was designed in this way and to make the study workflow easier to understand, it would be beneficial to use a visual abstract showing at which time-points, how many people in which groups (prior COVID infection/or not, etc.), and which tests were studied.

2. Could you explain the term 'intercept' and its purpose of usage in Table-1? What do the obtained values mean and how are they interpreted?

6. PLOS authors have the option to publish the peer review history of their article (what does this mean?). If published, this will include your full peer review and any attached files.

Reviewer #1: No

Reviewer #2: **Yes: **Yeşim Tuyji Tok, MD

---

## [Author Response · Author response to Decision Letter 0]

11 Jun 2023

Dear Dr. Harapan Harapan and Reviewers, 

Thank you for your consideration of this paper (PONE-D-23-08631) and the time you have taken to review it. We appreciate your thoughts and suggestions. Please see the responses below to see how we have incorporated your feedback into this new version. 

Journal Requirements:

Authors: Thank you for this reminder. We have ensured that both the manuscript and the file naming meet the journal style requirements. 

Authors: The funding statement has been amended to: "Funding for this study was provided by the National Institutes of Health (awards R01 AI150334 to RES). The analysis was supported by Global Health Equity Scholars and the National Institutes of Health (FIC D43TW010540 and T32AR007107 to RS). The funders had no role in the study design, data collection and analysis, decision to publish, or preparation of the manuscript.” It has also been included in the attached cover letter, thank you.

Important: If there are ethical or legal restrictions to sharing your data publicly, please explain these restrictions in detail. Please see our guidelines for more information on what we consider unacceptable restrictions to publicly sharing data: http://journals.plos.org/plosone/s/data-availability#loc-unacceptable-data-access-restrictions. Note that it is not acceptable for the authors to be the sole named individuals responsible for ensuring data access. We will update your Data Availability statement to reflect the information you provide in your cover letter.

Authors: We have shared our de-identified dataset as a supplemental excel file, “S1 File”. 

Authors: The de-identified data is being shared as a supplemental excel file, “S1 File” with this submission. 

Authors: Thank you for noting the difference in title (Vaccine vs vaccination). We have corrected the online submission form and the title is identical now at both places: “Correlation of a Commercial Platform’s Results with Post-vaccination SARS-CoV-2 Neutralizing Antibody Response and Clinical Host Factors”.

Authors: Thank you for this comment. We have updated the Methods section to include the following statement: “Approval for this study (SARS-CoV-2 Vaccine and You, Protocol #1599488) was obtained from the Veterans Affairs Connecticut Healthcare System (VACHS) Institutional Review Board (IRB) in December 2020. Written informed consent was obtained from willing participants recruited from COVID-vaccination clinics set up by VACHS beginning in December 2020.” 

Reviewers' comments:

Reviewer #1: This manuscript is an interesting topic, however, the data is not new and the content is suitable for publishing as a letter or short communication that extended the result from the previous publications. I suggest adding more data and text to make it suitable for the original article.

Authors: Thank you for this suggestion. We have added a Table 1 of demographics and comorbidities. We have incorporated more detail throughout our manuscript now, and feel that this made the manuscript clearer. We have elaborated on the ‘Methods’ section, including more details of statistical analysis. We redid the linear correlation to add an adjusted R-square value and tested it in several models to ensure it will be replicable in the field (linear model is in the body of the paper, the non-linear loess model is in the appendix. Although the loess model performed even better, it is harder to follow mathematically and less applicable to the clinical audience of PLOS, therefore we provide that in supplemental material). We also did a sensitivity analysis looking at the correlation if we removed any values that were above the limit of detection (read as >25,000 AU/mL) on the commercial platform we used. This sensitivity analysis demonstrated an improvement in the adjusted R-squared value of the correlation to 0.80. We can therefore conclude that the linear model we included in the body of this paper is the most conservative estimate of the true correlation between IgG and nAb IC50. This has been included in the ‘Results’ section. We thank the reviewer for encouraging us to add more data, and believe that this additional work has strengthened the conclusions while also providing more insight into the methodology.

Major concerns.

1. Please add the IRB/EC approval number to the Methods.

Authors: The ‘Methods’ section has been updated as noted above with the approved protocol number. 

2. Why you over emphasised the CMIA technique of the Abbott AdviseDx SARS-CoV-2 IgG II? The CMIA is only the technique, but the target of the antibody is more important than the technique of commercial diagnostic assays.

Suggest using "IgG anti-S1/RBD" or "IgG anti-RBD" instead of "IgG-II CMIA" to make it more specific.

See more: https://www.fda.gov/media/146371/download

Authors: We agree. It was not our intention to specifically highlight a single type of commercial test, but rather offer an example of how commercial IgG anti-S1/RBD testing may be used. We have updated the language throughout the manuscript with the exception of when describing the methodology used to measure the IgG anti-S1/RBD. Thank you for suggesting this.

3. Suggest creating a Demographic table including the clinical and COVID-19 vaccine used to delineate the participant's demographic data.

Authors: We have added a demographics table, named Table 1 as you suggested.

Minor concerns.

1. Methods, suggest being divided into subtitles, such as Participants, Immunologic assessment, and Statistical analysis.

Authors: Thank you for the suggestion, the manuscript has been updated with the ‘Methods’ section broken into the following categories: ‘Participants’, ‘Variable collection’, and ‘Statistical analysis’. 

2. Suggest adding the conversion factor of the Abbott AdviseDx to the introduction, the real result from this instrument report is in the AU/mL but BAU/mL is converted. Some readers may don't understand where the BAU/mL came from.

Authors: Thank you for this suggestion. We have added the AU/mL to BAU/mL conversion and interpretation factor to the ‘Methods’ section in the manuscript, which reads as follows. 

“IgG anti-S1/RBD were measured in AU/ml, a unit specific to the commercial platform used. One Abbott platform AU/ml converts to 7.04 BAU/ml, the standard World Health Organization (WHO) units used as a standard for commercial antibody platform result interpretation [8, 9].”

The following have been added as references: 

8. Müller L, Kannenberg J, Biemann R, Hönemann M, Ackermann G, Jassoy C. Comparison of the measured values of quantitative SARS-CoV-2 spike antibody assays. J Clin Virol. 2022;155:105269. doi:10.1016/j.jcv.2022.105269 

9. Perkmann T, Perkmann-Nagele N, Koller T, et al. Anti-Spike Protein Assays to Determine SARS-CoV-2 Antibody Levels: a Head-to-Head Comparison of Five Quantitative Assays. Microbiol Spectr. 2021;9(1):e0024721. doi:10.1128/Spectrum.00247-21

Our aim is to share this work with the field to illustrate that using a commercial platform can cost-effectively predict neutralizing ability of an individual’s serum against SARS-CoV-2. Since WHO has provided a standardization in BAU/mL, it helps the interpretation of results from various platforms, including the one we used, and adding this clarification strengthens the manuscript. We thank the reviewer for helping us specify this.

Comments.

1. Suggest using subscript of the "50" in the term "IC₅₀".

Authors: The formatting of IC50 is often investigator and journal dependent, and we have incorporated the suggested change now throughout the manuscript and figures.

Reviewer #2: Reviewer Suggestions

1. As it is known, antibody titers spontaneously decrease over time, and also may increase due to COVID-19 or vaccination. For this reason, to evaluate the correlations of the neutralizing antibody IC50 measurement and the anti-RBD-S1-IgG (IgG) antibody measurement, they must be studied from the sera samples taken at the same time-point. In order to indicate that the study was designed in this way and to make the study workflow easier to understand, it would be beneficial to use a visual abstract showing at which time-points, how many people in which groups (prior COVID infection/or not, etc.), and which tests were studied.

Authors: Thank you for this suggestion. Each sample was measured for IgG and IC50 at the same collection time point. We have clarified this in the ‘Methods’ section with the sentence: “Each measurement of IgG and IC50 was done on serum from the same collection time point to avoid any errors in measurement due to antibody fluctuation over time.” 

We have also added a supplemental visual abstract (S2 Fig) to illustrate how many samples at each time-point were from subjects that had COVID before that collection. We will defer to the journal to decide whether this new supplemental figure will be included in the publication. 

S2 Fig. Distribution of Subjects by Collection Time-point and COVID-status. COVID-positivity was defined as RT-PCR COVID- diagnosis at any time prior to collection. 

2. Could you explain the term 'intercept' and its purpose of usage in Table-1? What do the obtained values mean and how are they interpreted?

Authors: Thanks for this question. Each of the models in Table-1 (now Table-2) illustrates a formula that may be used to predict IC50. For example, Model 1 predicts IC50 from the variables IgG, COVID positive status, and renal function. With multiple datapoints, this model creates a line when graphed. The point at which that line intercepts the y axis, is the intercept.

---

## [Decision Letter · Decision Letter 1]

25 Jul 2023

Correlation of a Commercial Platform’s Results with Post-vaccination SARS-CoV-2 Neutralizing Antibody Response and Clinical Host Factors

PONE-D-23-08631R1

Dear Dr. Gupta,

We’re pleased to inform you that your manuscript has been judged scientifically suitable for publication and will be formally accepted for publication once it meets all outstanding technical requirements.

Kind regards,

Harapan Harapan, MD, PhD

Academic Editor

PLOS ONE

Additional Editor Comments (optional):

Reviewers' comments:

Reviewer's Responses to Questions

**Comments to the Author**

1. If the authors have adequately addressed your comments raised in a previous round of review and you feel that this manuscript is now acceptable for publication, you may indicate that here to bypass the “Comments to the Author” section, enter your conflict of interest statement in the “Confidential to Editor” section, and submit your "Accept" recommendation.

Reviewer #1: (No Response)

Reviewer #2: All comments have been addressed

2. Is the manuscript technically sound, and do the data support the conclusions?

Reviewer #1: Yes

Reviewer #2: Yes

3. Has the statistical analysis been performed appropriately and rigorously? 

Reviewer #1: Yes

Reviewer #2: Yes

4. Have the authors made all data underlying the findings in their manuscript fully available?

Reviewer #1: Yes

Reviewer #2: Yes

5. Is the manuscript presented in an intelligible fashion and written in standard English?

Reviewer #1: Yes

Reviewer #2: Yes

6. Review Comments to the Author

Reviewer #1: Thank you for your improvement in the quality and information in the revised manuscript. Have good luck.

Reviewer #2: Although the subject is out of date, it is a well-designed and valuable study. Comments have been adressed and explained sufficiently.

7. PLOS authors have the option to publish the peer review history of their article (what does this mean?). If published, this will include your full peer review and any attached files.

Reviewer #1: No

Reviewer #2: **Yes: **Yeşim Tuyji Tok

---

## [Editor Report · Acceptance letter]

21 Aug 2023

PONE-D-23-08631R1 

Correlation of a Commercial Platform’s Results with Post-vaccination SARS-CoV-2 Neutralizing Antibody Response and Clinical Host Factors 

Dear Dr. Gupta:

I'm pleased to inform you that your manuscript has been deemed suitable for publication in PLOS ONE. Congratulations! Your manuscript is now with our production department. 

Kind regards, 

on behalf of

Dr. Harapan Harapan 

Academic Editor

PLOS ONE